# Explainable detection of adverse drug reaction with imbalanced data distribution

**Jin Wang[1], Liang-Chih Yu[2]\*, Xuejie Zhang[1]**

**1** School of Information Science and Engineering, Yunnan University, Kunming, China, **2** Department of Information Management, Yuan Ze University, Taoyuan, Taiwan

\* lcyu@saturn.yzu.edu.tw

## Abstract

Analysis of health-related texts can be used to detect adverse drug reactions (ADR). The greatest challenge for ADR detection lies in imbalanced data distributions where words related to ADR symptoms are often minority classes. As a result, trained models tend to converge to a point that strongly biases towards the majority class and then ignores the minority class. Since the most used cross-entropy criteria is an approximation to accuracy, the model focuses more readily on the majority class to achieve high accuracy. To address this issue, existing methods apply either oversampling or down-sampling strategies to balance the data distribution and exploit the most difficult samples of the minority class. However, increasing or reducing the number of individual tokens alone in sequence labeling tasks will result in the loss of the syntactic relations of the sentence. This paper proposes a weighted variant of conditional random field (CRF) for data-imbalanced sequence labeling tasks. Such a weighting strategy can alleviate data distribution imbalances between majority and minority classes. Instead of using *softmax* in the output layer, the CRF can capture the relationship of labels between tokens. The locally interpretable model-agnostic explanations (LIME) algorithm was applied to investigate performance differences between models with and without the weighted loss function. Experimental results on two different ADR tasks show that the proposed model outperforms previously proposed sequence labeling methods.

## Author summary

Post-marketing drug safety surveillance offers the chance to detect serious ADRs resulting in hospitalization and ADRs occurring in patients, e.g., patients with high comorbidity and receiving drugs that are administered only in hospitals. This monitoring has traditionally been accomplished by surveying users. Recently, the automatically recording ADR of users in social media can greatly help biopharmaceutical enterprises to improve their products. Previous methods of name entity recognition in natural language processing were usually performed on the corpora with a balanced data distribution. Conversely, the datasets for ADR detection are extremely imbalanced. As a result, the detector tends to ignore the ADR symptoms and the related indications, which are more important. In

**Data Availability Statement:** The input data files and codes for model analysis and evaluation is available at: https://github.com/wangjin0818/adverse_drug_reaction/.

**Funding:** This work was supported by the National Natural Science Foundation of China (NSFC) under Grants Nos. 61702443 (to JW), 61966038 (to JW) and 61762091 (to XJZ), and in part by the Ministry of Science and Technology, Taiwan, ROC, under Grant No. MOST110-2628-E-155-002 (to LCY). The funders had no role in study design, data collection and analysis, decision to publish, or preparation of the manuscript.

**Competing interests:** The authors have declared that no competing interests exist.

this study, we propose a weighted CRF model based on BERT for the detection task of ADR. A weighted variant of the Viterbi Algorithm is implemented to assign more weight to the minority class, forcing the model to pay more attention to minority classes to ensure effective detection. The results suggested that the proposed method provides a significant performance boost without changing the model architecture in imbalanced-data tasks.

## Introduction

An adverse drug reaction refers to any injury caused by taking medication, and the incidence of such injuries is quite high especially in cases of large doses or long duration of medication use. However, such reactions are unpredictable, and pre-market clinical trials of new drugs are usually only conducted on samples of 500-3000 people, for a single type of disease, and often exclude special populations (e.g., the elderly, pregnant women, and children). Therefore, such trials often fail to identify relatively rare adverse reactions, late-onset reactions, or adverse reactions that occur in special populations, which only become apparent after following after large-scale use [1]. This raises an urgent need for post-marketing drug safety surveillance after drug approval [2–5].

This monitoring has traditionally been accomplished by surveying users, but more recent approaches have focused on automatic identification and extraction of ADR symptoms. These methods often apply sequence labeling techniques in natural language processing (NLP) to monitor social media interaction on platforms such as Twitter and Facebook. Traditional methods for sequence labeling use rule-based or statistic methods, such as support vector machine (SVM) [6], which are highly dependent on hand-crafted features, such as *n*-gram. Conditional random field (CRF) [7] is another commonly used statistical method for sequence labeling and ADR detection. Compared with classification models, such as SVM and neural network models with *softmax*, CRF can extract the dependencies between labels.

Recent advances in deep neural networks (DNN) [8–10] and representation learning [11, 12] have considerably improved the ability of algorithms to process text. Building neural networks, such as convolutional neural networks (CNN) [13], recurrent neural networks (RNN) [14], long short-term memory (LSTM) [15] and BiLSTM-CRF [16, 17] can be an effective approach for in-depth research. Furthermore, attention mechanisms can be applied to improve the performance of DNN models to extract more task-specific features between tokens to provide meaningful information [18]. Other effective approaches apply the pre-trained language models (PLM), such as BERT [19], RoBERTa [20] and ALBERT [21], to provide powerful representation to boost the performance of sequence labeling.

One of the biggest stumbling blocks in ADR tasks is data distribution, which often appears in conventional sequence labeling tasks and corpora. As shown in Table 1, the words of ADR symptoms in both examples from social media only take respective ratios of 6.23% and 8.3%. Table 2 presents concrete examples of several sequence labeling tasks, in which most tokens are annotated as class O, which is about 10 times that of ADR with entity labels. The algorithms tend to produce unsatisfactory classifiers when faced with (even extremely) imbalanced datasets. Those models may have a bias towards classes and only predict the majority class.

This is because the frequently used loss function in most text mining tasks is categorical cross-entropy, which approximates the accuracy score. Unfortunately, this is not a suitable metric for imbalanced-data circumstances since the minority class has less effect on accuracy than the majority class [22]. For instance, in a ratio of 99:1 between the majority and minority

**Table 1. The imbalance examples in labeling of adverse drug reactions.** The words of ADR symptoms in both examples from social media only take respective ratios of 6.23% and 8.3%.

| Example 1: | |
|---|---|
| Text | $Thanks_1$ $god_2$ $my_3$ $steady_4$ $stream_5$ $of_6$ $Vyvanse_7$ $is_8$ $wearing_9$ $off_{10,11}$ $I_{12}$ $can_{13}$ $sleep_{14}$ $now_{15.16}$ |
| Label | $O_1$ $O_2$ $O_3$ $O_4$ $O_5$ $O_6$ $O_7$ $O_8$ $O_9$ $O_{10}$ $O_{11}$ $O_{12}$ $O_{13}$ $B\text{-}ADR_{14}$ $O_{15}$ $O_{16}$ |
| Ratio | 6.25% |
| Example 2: | |
| Text | $When_1$ $the_2$ $depression_3$ $is_4$ $coming_5$ $out_6$ $caused_7$ $by_8$ $this_9$ $damn_{10}$ $levofloxacin_{11}$ $:)_{12}$ |
| Label | $O_1$ $O_2$ $B\text{-}ADR_3$ $O_4$ $O_5$ $O_6$ $O_7$ $O_8$ $O_9$ $O_{10}$ $O_{11}$ $O_{12}$ |
| Ratio | 8.3% |

classes, the trained model would classify everything to the majority class, since doing so can achieve 99% accuracy. However, it will be useless for tasks where the minority classes are more important than the majority class. In most circumstances, false negatives can have higher importance, while false positives are of course undesirable. As a result, the important minority classes, i.e., ADRs and indications, will be completely ignored by such a classifier.

In this paper, we propose a weighted pre-trained language model which is both robust to the strong class imbalance and able to integrate dependencies and syntactical information between tokens. This model implements a weighted variant of the CRF loss function. By assigning more weights to the minority class, the model can be forced to pay more attention to such classes, for effective detection. Furthermore, we introduce an explainable algorithm which provides a qualitative understanding between the input features and the corresponding prediction to compare the behavior of models with and without the weighted loss function.

Experiments were conducted on several strongly class-imbalanced ADR corpora. The proposed model is compared against several current sequence labeling models. The results show that the proposed model provides a better solution to handle the imbalanced-data issue, thus improving performance. In addition, both visualizations and qualitative results are presented to demonstrate the effect of the weighted loss function on CRF.

# Results

This section presents the experiments conducted on several corpora to evaluate the performance of the proposed weighted BERT-CRF model against different neural networks for the ADR detection task.

## Datasets

To evaluated the effectiveness of the proposed weighted BERT-CRF model, the comparative experiments were conducted on two corpora. Notably, the split strategies for train and test sets

**Table 2. Statistic results on the imbalance datasets of adverse drug reactions.** Most tokens are annotated as class O, which is about 10 times that of ADR with entity labels. The algorithms tend to produce unsatisfactory classifiers when faced with (even extremely) imbalanced datasets. Those models may have a bias towards classes and only predict the majority class.

| Datasets | Samples | Max length | Mean length | Vocab | Tokens | O | | ADR | | Indication | |
|---|---|---|---|---|---|---|---|---|---|---|---|
| | | | | | | Tokens | Ratio | Tokens | Ratio | Tokens | Ratio |
| Twitter | 844 | 36 | 19.0 | 2,843 | 16,023 | 13,852 | 86.5% | 1,970 | 12.3% | 201 | 1.3% |
| PubMed | 4,858 | 93 | 21.3 | 7,950 | 103.302 | 89,331 | 86.5% | 13,971 | 13.5% | - | - |

were different, for a fair comparison with the performance that were reported in the previous studies.

- **Twitter** consists of two datasets, including *Twitter ADR datasets* (v1.0) [7, 23] and *ADHD* datasets [15]. By collecting and annotating user published tweets, the *Twitter ADR datasets* (v1.0) contain references to 81 drugs and newly reported terms which can be commonly found in the US market. *ADHD* is a supplement of Twitter including at least one ADR or indication label. According to Cocos et al. [15], the datasets were split into a training set and a testing set with an 3:1 ratio. For hyper-parameter fine-tuning, we also split a dev set from the training set with a ratio of 8:2.

- **PubMed** abstracts [24] is a biomedical text dataset, in which each sentence contains at least one ADR label. Following a previous study [25], we slightly modified the datasets as follows: 1) ensure that each sentence contains only one related-drug reference and a list of corresponding ADRs caused by the drug; 2) remove 120 sentences which contain the name of drug which are commonly regarded as the cause of adverse reactions, e.g., *theophylline poisoning*, where *theophylline* is the cause of *poisoning*. In addition, the dataset was split into training/dev/test set with a ratio of 8: 1: 1.

Each sample in both two datasets contains at least one ADR or indication label mentioned. Details of above corpora are summarized in Table 2.

## Evaluations metrics

The performance of ADR detection is evaluated by using approximate matching [24]. For example, given a tweet *the Seroquel gave me lasting sleep paralysis with true ADR labels sleep paralysis*, the prediction of either *lasting sleep paralysis* or simply *paralysis* are both regarded as correct. The corresponding metrics are precision, recall and $F_1$-score, defined as follows

$$Precision = \frac{TP_{ADR}}{TP_{ADR} + FP_{ADR}} \tag{1}$$

$$Recall = \frac{TP_{ADR}}{TP_{ADR} + FN_{ADR}} \tag{2}$$

$$F_1-score = \frac{2 \times Precision \times Recall}{Precsion + Recall} \tag{3}$$

where $TP_{ADR}$ is the number of the ADR tokens that are approximately matched, and $TP_{ADR} + FP_{ADR}$ and $TP_{ADR} + FN_{ADR}$ are respectively the number of all predicted and real ADR tokens. A higher $F_1$-score indicate better detection performance.

## Baselines

To evaluate the proposed weighted BERT-CRF model, we implement several baselines for comparison. The details are presented as follows.

**Context encoder.**   For special tasks, different representations may impact the final performance. Therefore, we implement different context encoders to produce representations for the ADR detection tasks.

- **BiLSTM** [16, 17] is an effective conventional approach for sequence labeling tasks. It encodes word embeddings into hidden representations for sequence labeling. To address out-of-vocabulary (OOV) issues, we also introduce char embeddings (**BiLSTM-char**) [26].

Each word representation is composed of the embeddings of its constituent characters. To further improve performance, self-attention was introduced after BiLSTM to align the target token to its contexts (**BiLSTM-attention**).

- **ELMo** [27] is a pre-trained language model based on LSTM, which can be transferred for NLP tasks. It can be extended by adding a BiLSTM layer (**ELMo-BiLSTM**) to learn long-distance dependencies.

- **BERT** [19] can be used as fixed input features for the following labeling tasks (**BERT-fix**). In addition, it can be transferred and fine-tuned for a wide range of tasks by adding a classifier without any task-specific architectural modification (**BERT-finetune**). To capture long distance dependency, a BiLSTM was added for both two models.

**Classifiers.** Both *softmax* and CRF were used as the output layers to label tokens in the sequence. In addition, a weighted variant of *softmax* (wSoftmax) and the proposed weighted CRF (wCRF) were also implemented for comparison, to further investigate whether the weighting mechanism can improve performance.

Hyper-parameters are fine-tuned on the corresponding development set of each dataset. For BiLSTM, the word vectors were pre-trained using GloVe on the 840B Common Crawl corpus [28]. The dimensionality of the word vectors is 300. Words that don't appear in GloVe were initialized with a uniform distribution $U(-0.25, 0.25)$. Char embeddings are initialized randomly with a dimensionality of 50, and then updated along with model training. To avoid overfitting, we apply a dropout layer after the context encoder layer, and the dropout rate was set to 0.1 for all datasets. Pre-trained language models, i.e., ELMo and BERT, are initialized from a checkpoint. For ELMo and BERT, `Original (5.5B)` and `BERT-base-uncased` pretrained model were respectively used. The Adam optimizer was adopted to train all models.

## Hyper-parameters fine-tuning

Several parameters may impact the final performance of the proposed weighted BERT-CRF model on both the **Twitter** and **PubMed** datasets. Fig 1 shows the hyper-parameter fine-tuning to search for optimal settings according to the final performance on the development set using the grid-search strategy. That is, we fine-tune each parameter for the optimal value in turn. When the optimal value of one parameter is obtained, it will be fixed, and the next parameter will be fine-tuned. As indicated, for **Twitter** and **PubMed**, the best performance is achieved when the number of training epochs are respectively 9 and 12, the batch size is 16 and 16, learning rate is 2e-5 and 2e-5, and the dimensionality of the hidden state in BiLSTM is 256 and 256. Once these parameters exceed the optimal settings, the $F_1$-score declines. The results indicate that appropriate parameters can ensure that the model obtains the interactive relationship and syntactic information even when the distribution is imbalanced, thus improving classification performance.

Once the optimal parameters settings are obtained, they are used for token labeling on the test sets for both **Twitter** and **PubMed** corpora. Comparing the results on the development set and that on the test set in Table 3 shows that the performance on the test set was very close to that on the development set for both ADR datasets.

## Comparative results

Table 3 shows the experimental results on both the **Twitter** and **PubMed** datasets. We further combined different encoders and classifiers for ADR detection. For conventional models, BiLSTM-Attention assigns different weights to the contexts according to the contribution to

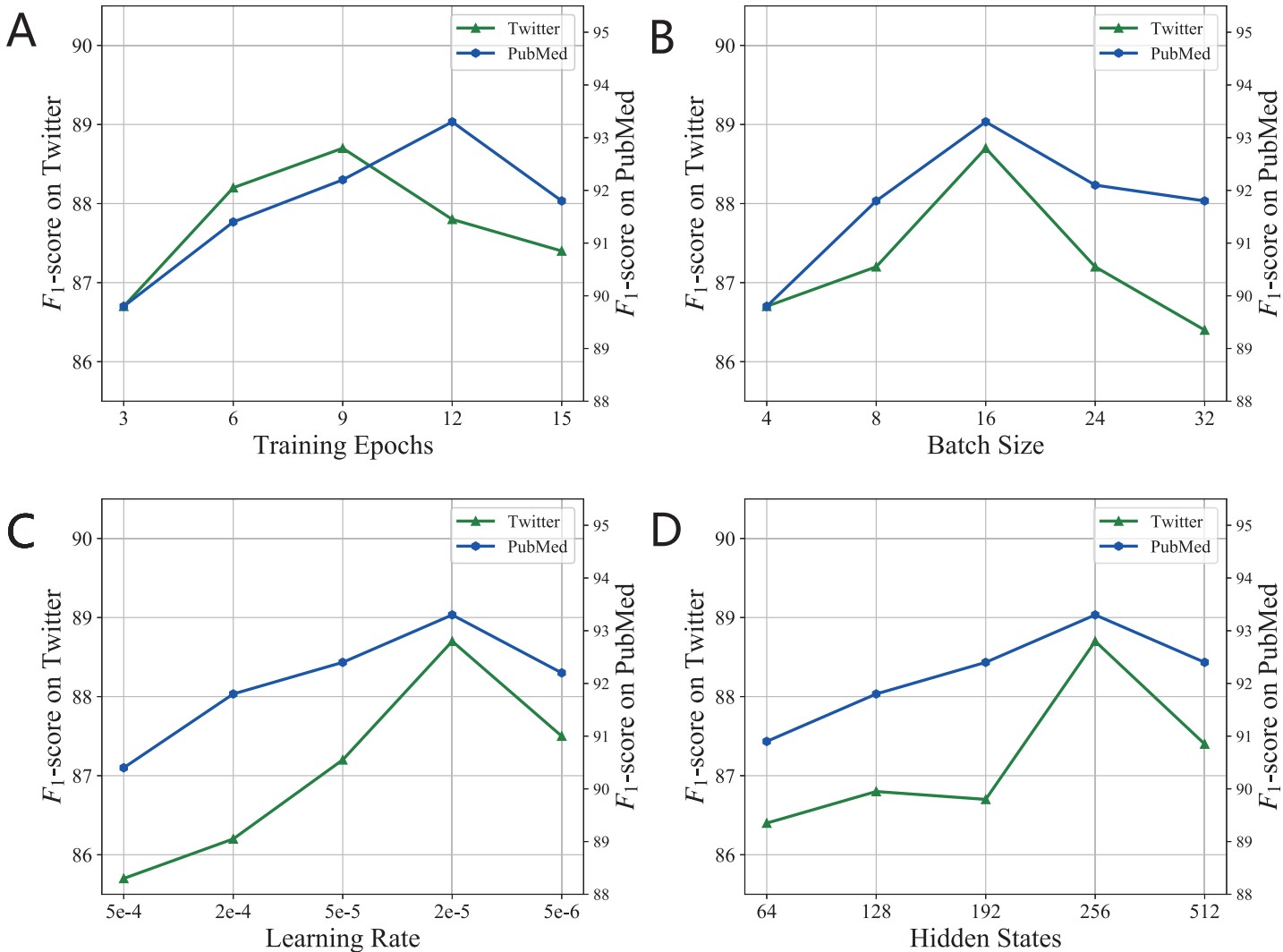

**Fig 1. Hyper-parameters finetuning for the proposed weighted BERT-CRF model.** (A) The best performance is achieved when the number of training epochs are respectively 9 and 12. (B) The optimal settings for the batch size is 16 and 16. (C) The optimal learning rate is 2e-5 and 2e-5. (D) The dimensionality of the hidden state in BiLSTM is 256 and 256.

the final classification of the target token, thus it outperformed BiLSTM and BiLSTM-char by about 1.7% and 1.2%. PLMs, such as ELMo and BERT, can further improve performance on BiLSTM. PLM can provide better contextual representations than conventional word vectors e.g., GloVe, thus yielding better performance. Instead of using a fixed contextual representation for tokens, i.e., ELMo and BERT-fix, BERT was also fine-tuned in the training procedure for the entire model, thus achieving the best performance on both the **Twitter** and **PubMed** datasets. In addition, the weighting strategy on both *softmax* and CRF can alleviate the imbalanced data distribution, and they thus outperformed their conventional versions by about 1.1% and 1.8% on average across the two ADR tasks.

For the output layer, recent studies suggest using the *softmax* function for token labeling, which performs well in independent prediction. However, its efficiency decreases when there is a strong dependency between labels. The grammar rules in the sequence labeling task limit

**Table 3. Results comparison of different context encoders w/ and w/o weighted mechanism for ADR detection tasks.** The proposed weighted CRF significantly outperformed several baselines on both the Twitter and PubMed datasets. In addition, the weighting strategy on both *softmax* and CRF can alleviate the imbalanced data distribution, and they thus outperformed their conventional versions by about 1.1% and 1.8% on average across the two ADR tasks.

| Model | | Twitter | | | PubMed | | |
|---|---|---|---|---|---|---|---|
| | | Precision | Recall | $F_1$-score | Precision | Recall | $F_1$-score |
| BiLSTM | Softmax | 78.6 | 82.6 | 80.5 | 88.1 | 87.3 | 87.6 |
| | CRF | 77.1 | 84.7 | 80.7 | 89.0 | 87.6 | 88.3 |
| | wSoftmax | 77.2 | 84.6 | 80.7 | 89.6 | 87.9 | 88.7 |
| | wCRF | 77.2 | 88.4 | 82.4 | 89.9 | 88.1 | 89.0 |
| BiLSTM-char | Softmax | 76.9 | 80.1 | 78.5 | 87.5 | 86.7 | 87.1 |
| | CRF | 77.0 | 82.1 | 79.4 | 88.2 | 87.7 | 88.0 |
| | wSoftmax | 76.2 | 85.1 | 80.4 | 87.5 | 89.2 | 88.3 |
| | wCRF | 78.0 | 84.7 | 81.2 | 89.3 | 90.1 | 89.7 |
| BiLSTM-Attention | Softmax | 78.5 | 83.5 | 80.9 | 88.0 | 86.8 | 87.4 |
| | CRF | 79.2 | 85.8 | 82.8 | 87.5 | 88.6 | 88.0 |
| | wSoftmax | 77.0 | 86.3 | 81.4 | 87.2 | 89.6 | 88.5 |
| | wCRF | 79.9 | 87.7 | 83.6 | 91.5 | 91.8 | 91.7 |
| ELMo | Softmax | 78.1 | 87.8 | 82.7 | 90.2 | 88.7 | 89.4 |
| | CRF | 79.9 | 86.8 | 83.2 | 90.6 | 89.6 | 90.1 |
| | wSoftmax | 80.5 | 86.6 | 83.4 | 90.8 | 90.0 | 90.4 |
| | wCRF | 81.8 | 88.0 | 84.8 | 91.1 | 90.4 | 90.7 |
| ELMo-BiLSTM | Softmax | 79.3 | 86.8 | 83.2 | 88.3 | 90.1 | 89.2 |
| | CRF | 79.9 | 87.5 | 83.5 | 91.0 | 90.6 | 90.8 |
| | wSoftmax | 79.9 | 89.2 | 84.3 | 90.5 | 91.5 | 91.1 |
| | wCRF | 79.4 | 90.4 | 85.6 | 91.9 | 92.6 | 92.4 |
| BERT-fix | Softmax | 79.5 | 88.0 | 83.5 | 91.9 | 91.8 | 91.9 |
| | CRF | 81.8 | 88.0 | 84.8 | 92.4 | 92.0 | 92.2 |
| | wSoftmax | 82.1 | 88.6 | 85.2 | 92.4 | 92.1 | 92.3 |
| | wCRF | 83.8 | 89.9 | 86.8 | 93.1 | 92.4 | 92.8 |
| BERT-finetune | Softmax | 82.5 | 90.7 | 86.4 | 91.5 | 91.8 | 91.6 |
| | CRF | 82.9 | 90.7 | 86.6 | 91.9 | 92.8 | 92.2 |
| | wSoftmax | 82.5 | 91.6 | 86.8 | 90.8 | 92.9 | 91.8 |
| | wCRF | **85.4** | **92.4** | **88.7** | **93.6** | **93.2** | **93.3** |

the label between tokens that cannot be predicted independently. For example, $<I\text{-}ADR>$ cannot be connected after $<B\text{-}Indication>$, or after the token that starts with the position. Using CRF can address this problem. For each generated label sequence, CRF uses a score to represent the sequence quality, with a higher score indicating that the currently generated label sequence performs better. The results demonstrate that the proposed BERT-CRF benefits from both contextual representation of BERT and weighted loss function of CRF, thus it can achieve the best performance for ADR detection.

Table 4 shows the comparative performance of the proposed BERT-CRF against the state-of-the-art models on both the **Twitter** and **PubMed** datasets. As indicated, the proposed model outperformed these models by 5.1% and 3.0%, respectively.

Further, the ability of BERT to handle the sequential encoding mainly depends on position embeddings, which are added with token embeddings and segment embeddings as inputs to the BERT model. With the increase of the number of layers, the relevant sequence information will slightly dissipate in the output layer. For the conventional sequence labeling task, the use of LSTM followed by the BERT encoder can actually enhance the sequential information.

**Table 4. Comparative results of the proposed weighted BERT-CRF model against the previously proposed model.** The proposed model outperformed the state-of-the-art models on both Twitter and PubMed datasets by 5.1% and 3.0%, respectively.

| | Twitter | | | PubMed | | |
|---|---|---|---|---|---|---|
| | **Precision** | **Recall** | **$F_1$-score** | **Precision** | **Recall** | **$F_1$-score** |
| Cocos et al. [15] | 70.4 | 82.9 | 75.5 | - | - | - |
| Ramamoorthy et al. [25] | - | - | - | 88.4 | 82.4 | 85.3 |
| Ding et al. [18] | 78.5 | 91.4 | 84.4 | 86.7 | 94.8 | 90.6 |
| Weighted BERT-CRF | 85.4 | 92.4 | 88.7 | 93.6 | 93.2 | 93.3 |
| Weighted BERT-CRF w/o BiLSTM | 83.4 | 91.8 | 87.4 | 91.8 | 91.4 | 91.6 |

Especially when the output classifier is CRF, such a combination can provide a better context information to support the implementation of CRF. To evaluate the effect of BiLSTM, an ablation study was conducted in Table 4. The result show that the removal of BiLSTM will degrade the performance of the model.

## Discussion

This section used an explainable algorithm to present some explanation or speculation on the empirical results.

### The effect of different loss function

The performance of the proposed weighted CRF mainly depends on the weight assignment strategies. That is, the loss of the minority classes should be assigned a heavier weight than that of the minority classes. For comparison, we introduce two other strategies. The first one uses the inverse value of the sample numbers (**Strategy-1**), and the other uses the inverse ratio of the sample numbers (**Strategy-2**). As indicated in Fig 2A, the proposed weights assignment strategy in Eq 12) (**Weighted Loss**) outperformed both of the aforementioned strategies, since the proposed method considers both the number of samples and the ratio of each class.

In addition, recent studies recommended using either focal loss [29] or dice loss [30] for multi-label classification with imbalanced data distributions. The focal loss can reduce the weights of the samples of the majority classes, and force the model to focus more on the samples that are difficult to detect during training. Unlike cross-entropy, dice loss was designed to fit an approximation of $F_1$-score metric to attach similar importance to the samples of the minority classes. As shown in Fig 2B, the proposed weighted loss achieved better performance than both loss functions. Since both focal loss and dice loss use a *softmax* function as the output layer, the strong dependency between labels will be ignored.

### Interpretability analysis

In text classification, the classifier is often a black-box tool, where the internal working mechanisms are completely hidden from the user. By using the deep learning models, the user may not know which features are most important for the final prediction. To further explore the effectiveness of the proposed weighted BERT-CRF model, Fig 3 randomly selects two examples of the test samples from both the **Twitter** and **PubMed** datasets and visualizes the contribution of the contexts for the target token.

To understand the prediction of the ADR detection model for a certain sample, the desired explanation should be local, i.e., must correspond to how the model behaves in the neighborhood of the token being predicted. Thus, the LIME algorithm was introduced to provide a qualitative understanding of the relationship between the input tokens and the corresponding

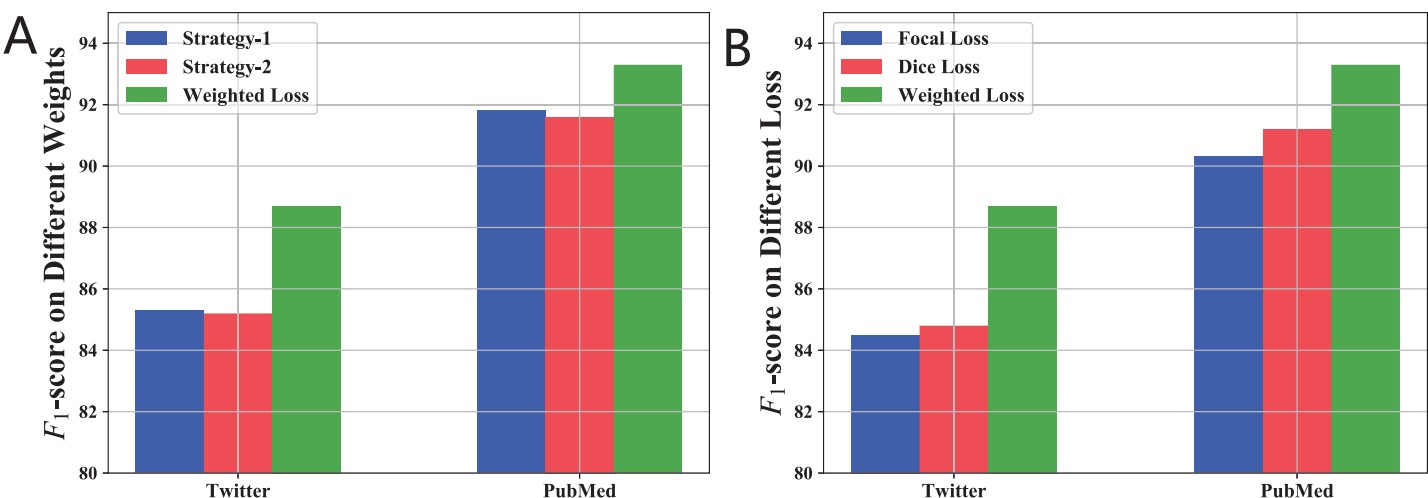

**Fig 2. Comparative results of the proposed weighted BERT-CRF model with different weights assignment and loss function.** (A) Different weight assignment. The performance of the proposed weighted CRF mainly depends on the weight assignment strategies. The green bar shows the proposed weight assignment (**Weighted Loss**), as described in Eq 12. For comparison, we introduce two other strategies. The blue bar shows the inverse value of the sample numbers (**Strategy-1**), and the red bar shows the inverse ratio of the sample numbers (**Strategy-2**). (B) Different loss function. Recent studies recommended using either focal loss or dice loss for multi-label classification with imbalanced data distribution. The green bar shows the performance of the proposed weighted loss function (**Weighted Loss**). The blue bar shows the performance of the focal loss, which can reduce the weights of the samples of the majority classes, and force the model to focus more on the samples that are difficult to detect during training. The red bar shows the performance of the dice loss, which was designed to fit an approximation of $F_1$-score metric to attach similar importance to the samples of the minority classes.

labels. To use the LIME algorithm, we rephrase the ADR detection task as a simple multiclass classification problem. To explain the effectiveness of the weighting strategy, the LIME algorithm was applied to explain the classification of tokens with ADR labels.

As indicated, green and red respectively means that the portion contributed positively and negatively to the classification of the target label. The weights are interpreted by applying them to the prediction probabilities. For the first example, if tokens *go* and *bed* were removed from the texts, the classifier is expected to predict *tired* as *<B-ADR>* with a probability $0.95 - 0.33 - 0.31 = 0.31$. Thus, the tokens *go* and *bed* could be regarded as indicators of ADR. Compared with the model without a weighting strategy, the proposed model can accurately predict the ADR label based on the local information.

For the second example, the proposed model predicted *<B-ADR>* and *<I-ADR>* for the tokens *gain* and *weight*. The word *pristiq* is a strong indicator that those tokens are ADR. This indicates that, in the dataset, *pristiq* is often a drug which may cause an ADR. In contrast, the model without the weighted loss function tends to ignore both the *<B-ADR>* and *<I-ADR>* label for the tokens *gain* and *weight*, even though these is a strong indicator *pristiq*. Since the model applies cross-entropy as loss function, it tends to predict all *<O>* labels for all tokens to achieve the lowest entropy value.

## Methods

Fig 4 shows the overall framework of the proposed weighted BERT-CRF model for ADR, which consists of three parts. The first part is a pre-trained BERT model, the second part is a bi-directional LSTM and the third part is a weighted CRF output layer. The details of each part are described as follow.

**A**

y=B-ADR (probability 0.9567, score 5.826) top features

| Contribution? | Feature |
| --- | --- |
| +5.929 | Highlighted in text (sum) |
| -0.103 | <BIAS> |

the day after my humira jab and as ever i'm mega tired . may have to go back to bed .

y=I-ADR (probability 0.0326, score -5.504) top features

| Contribution? | Feature |
| --- | --- |
| -0.342 | <BIAS> |
| -5.162 | Highlighted in text (sum) |

the day after my humira jab and as ever i'm mega tired . may have to go back to bed .

y=O (probability 0.0107, score -8.503) top features

| Contribution? | Feature |
| --- | --- |
| -0.210 | <BIAS> |
| -8.503 | Highlighted in text (sum) |

the day after my humira jab and as ever i'm mega tired . may have to go back to bed .

**B**

y=I-ADR (probability 0.963, score 3.679) top features

| Contribution? | Feature |
| --- | --- |
| +4.076 | Highlighted in text (sum) |
| -0.398 | <BIAS> |

my sleep schedule is all fucked up and pristiq is making me gain weight and i don't like this .

y=B-ADR (probability 0.0014, score -4.229) top features

| Contribution? | Feature |
| --- | --- |
| -1.924 | <BIAS> |
| -2.306 | Highlighted in text (sum) |

my sleep schedule is all fucked up and pristiq is making me gain weight and i don't like this .

y=O (probability 0.0107, score -8.503) top features

| Contribution? | Feature |
| --- | --- |
| -0.762 | <BIAS> |
| -4.208 | Highlighted in text (sum) |

my sleep schedule is all fucked up and pristiq is making me gain weight and i don't like this .

**Fig 3. Interpretability analysis of the selected examples for the proposed weighted BERT-CRF model.** Green and red respectively means that the portion contributed positively and negatively to the classification of the target label. The weights are interpreted by applying them to the prediction probabilities. (A) Example 1 (target = *tired*). If tokens *go* and *bed* were removed from the texts, the classifier is expected to predict *tired* as <B-ADR> with a probability 0.95 − 0.33 − 0.31 = 0.31. Thus, the tokens *go* and *bed* could be regarded as indicators of ADR. Compared with the model without a weighting strategy, the proposed model can accurately predict the ADR label based on the local information. (B) Example 2 (target = *weight*). The proposed model predicted <B-ADR> and <I-ADR> for the tokens *gain* and *weight*. The word *pristiq* is a strong indicator that those tokens are ADR. This indicates that, in the dataset, *pristiq* is often a drug which may cause an ADR. In contrast, the model without the weighted loss function tends to ignore both the <B-ADR> and <I-ADR> label for the tokens *gain* and *weight*, even though these is a strong indicator *pristiq*. Since the model applies cross-entropy as loss function, it tends to predict all <O> labels for all tokens to achieve the lowest entropy value.

## Context encoder

The context encoder module in our model is based on BERT, which achieved impressive performance in various NLP tasks. It consists of multilayers of bidirectional transformer encoders [31], and is pre-trained by unsupervised learning of either masked language model (with a masked ratio of 15%) or next sentence prediction. The uncased BERT-base model was used, containing 12 layers of transforms with a hidden size of 768.

We use the sentences from one input sample $\mathbf{X}$ as the context input, which can be divided and transformed as a sequence of token embeddings, denoted as $\mathbf{X} = [x_1, x_2, \cdots, x_N]$, where $N$ is the number of tokens in this sample. The model is required to predict the label sequence $\mathbf{y} = [y_1, y_2, \cdots, y_N]$ for the tokens. By using the WordPiece [32] tokenizer, a word may be tokenized as several subwords. For instance, the ADR symptom *depression* can be tokenized into *de*,

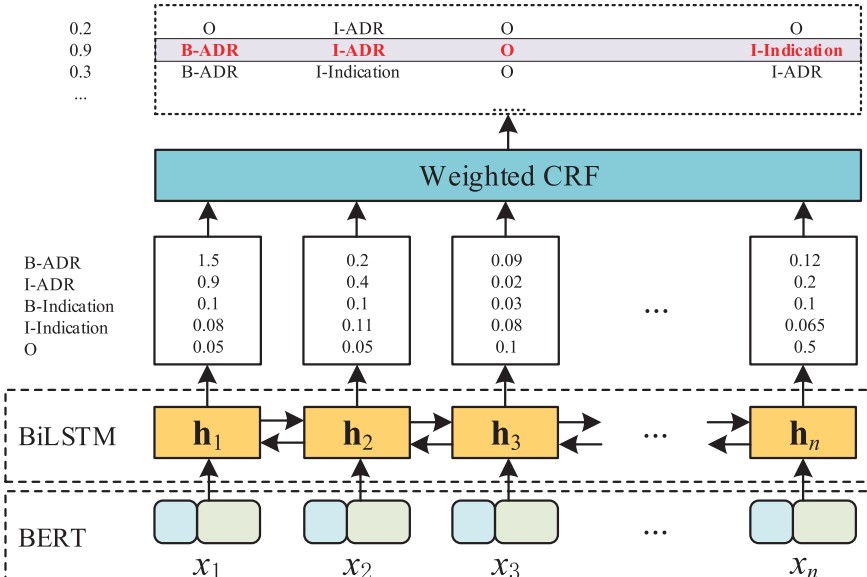

**Fig 4. System architecture of weighted BERT-CRF model.** It consists of three parts. The first part is a pre-trained BERT model, the second part is a bi-directional LSTM and the third part is a weighted CRF output layer.

*##press* and *##ion*, where the symbol ## means the token is not at the front of a word. In the implementation, the BOI tagging scheme was introduced to each label, and the model was trained only on the tag labels for the first subword of a split token. If the label for *depression* is <*B-ADR*>, only *de* will be set to <*B-ADR*> while the labels *##press* and *##ion* will be set to a special ignored token [IGN].

Using BERT, we add a special symbol $x_0$, i.e. [CLS], in front of each input sample. By concatenating with both position embeddings [33] and segmentation embeddings, the token embeddings were fed into the BERT model to get the output representation, $\mathbf{t}_i \in \Re^{d_t}$, denoted as,

$$[\mathbf{t}_0, \mathbf{t}_1, \ldots, \mathbf{t}_N] = \text{BERT}([x_0, x_1, \ldots, x_N]; \theta_{\text{BERT}}) \qquad (4)$$

where $\theta_{\text{BERT}}$ is the trainable parameters of the BERT model which is fine-tuned during model training, and $d_t$ = 768 is the dimensionality of the local representation. The contextualized sentence-level representation $[\mathbf{t}_1, \mathbf{t}_2, \cdots, \mathbf{t}_N]$ are used as the input embeddings of bi-directional LSTM layer, denoted as,

$$[\mathbf{h}_1, \mathbf{h}_2, \ldots, \mathbf{h}_N] = \text{BiLSTM}([\mathbf{t}_1, \mathbf{t}_2, \ldots, \mathbf{t}_N]; \theta_{\text{BiLSTM}}) \qquad (5)$$

where $\theta_{\text{BiLSTM}}$ is the corresponding trainable parameters of the BiLSTM model. Notably, BERT was fine-tuned in the training phase of the whole model.

## Conditional random fields

CRF is a type of undirected discriminative graph model [23] defined as a Markov random field. Since CRF outperforms *softmax* in capturing the relationship between neighboring information, it is selected to encode known relationships between the hidden representation $\mathbf{H} = [\mathbf{h}_1, \mathbf{h}_2, \cdots, \mathbf{h}_N]$ in the BiLSTM layer and the output label for each token in sentence. For a

sequence of label,

$$\mathbf{y} = [y_1, y_2, \ldots, y_N] \tag{6}$$

we define the score of the $i$-th token as,

$$s(x_i, y_i) = A_{y_{i-1}, y_i} + P_{i, y_i} \tag{7}$$

where $\mathbf{A}$ is a transition matrix, and its element $A_{i,j}$ represents the score of a transition from label $i$ to label $j$. We manually add the start and end labels of a sentence as $y_0$ and $y_{N+1}$, which is also added to the set of possible labels. $\mathbf{A}$ is therefore a square matrix of size $K + 2$. It can be inferred from Eq 7 that the total score of the sequence is equal to the sum of the score of each token, which is mainly decided by two parts. One is the emission matrix $\mathbf{P}$ which is the output $\mathbf{H}$ by the previous BiLSTM layer, and the other is the transition matrix $\mathbf{A}$ which is associated with CRF layer. Then, a *softmax* over all possible label sequence yields a probability for the sequence $\mathbf{y}$,

$$p(\mathbf{y}|\mathbf{X}) = \frac{\exp(s(\mathbf{X}, \mathbf{y}))}{\sum_{\mathbf{y}' \in \mathbf{Y_X}} \exp(s(\mathbf{X}, \mathbf{y}'))} \tag{8}$$

The CRF can be trained by maximizing the log-likelihood, defined as,

$$
\begin{aligned}
&\max \ \log(p(\mathbf{y}|\mathbf{X})) \\
&= \max \ s(\mathbf{X}, \mathbf{y}) - \log\left(\sum_{\mathbf{y}' \in \mathbf{Y_X}} \exp(s(\mathbf{X}, \mathbf{y}'))\right) \\
&= \min - \left(s(\mathbf{X}, \mathbf{y}) - \operatorname*{logadd}_{\mathbf{y}' \in \mathbf{Y_X}} \exp(s(\mathbf{X}, \mathbf{y}'))\right)
\end{aligned}
\tag{9}
$$

where $\mathbf{Y_X}$ denotes the possible path to all labels for the input sentence $\mathbf{X}$. It is evident that we encourage our network to produce a valid sequence of output labels. The item log add $s(\mathbf{X}, \mathbf{y}')$ is defined as a summation over all possible label sequences. The computation cost will be $O(K^N)$, where the number of possible label sequences grows exponentially with the sequence length. Here, we apply a dynamic programming method, the forward-backward algorithm, to solve it effectively and reduce the computation cost to $O(NK^2)$.

## Weighted loss function

The idea of the imbalanced learning approach is to design a weighted loss function that includes a distinct cost for each class. The main difficulty in Eq 10 is writing the last logadd item as a sum over observations. In the context of the hidden Markov model (HMM), Rabiner [34] has shown that the computation of a logadd item can be rewritten as a product over observations as,

$$\operatorname*{logadd}_{\mathbf{y}' \in \mathbf{Y_X}} \exp(s(\mathbf{X}, \mathbf{y}')) = \sum_{i=1}^{n} \log \exp_{\mathbf{y}' \in \mathbf{Y}_i} s(\mathbf{X}, \mathbf{y}') \tag{10}$$

Therefore, the weighted CRF objective function becomes,

$$\max \; \log(p(\mathbf{y}|\mathbf{X}))$$

$$= \min - \sum_{k=1}^{K} w_k \sum_{\{i|y=k\}} \left( \sum_{i=0}^{n} A_{y_{i-1}, y_i} \right. \tag{11}$$

$$\left. + \sum_{i=1}^{n} P_{i, y_i} - \log \exp_{\mathbf{y}' \in Y_i^k} s(\mathbf{X}, \mathbf{y}') \right)$$

where $\mathbf{Y_i^k}$ is all possible paths to the label of the $k$-th class for the $i$-th token, and $w_k$ is the cost parameter to weight the terms associated with the $k$-th class, given by,

$$w_k = \frac{N}{K \times n_k} \tag{12}$$

where $N$ is the number of all tokens, $k$ is the total number of classes, and $n_k$ is the number of tokens in the $k$-th class.

## Related works

An adverse drug reaction is an injury resulting from the use of a drug. People suffering from such reactions frequently recount their experiences in social media texts. This section presents a brief review of existing methods for detecting ADR by using sequence labeling techniques.

### Adverse drug reaction detection

Existing methods for ADR detection can be broadly divided into two categories: conventional models and deep neural networks. ADRMine [2] manually transforms the input texts and augments them with additional features, such as ADR lexicons, word contexts, cluster features of word embeddings and part-of-speech (POS) tags, in order to increase the effectiveness of the resulting CRF model [35]. Here, CRFs are a class of statistical modeling methods often applied in machine learning and used for structured prediction. Whereas a classifier predicts a label for a single sample without considering *neighbouring* samples, a CRF can take context into account. To do so, the predictions are modelled as a graphical model, which represents the presence of dependencies between the predictions. Unfortunately, Such a model heavily depend on feature engineering, which is time consuming and label intensive.

Recent studies have shown that deep neural network models perform well on several sequence labeling tasks. Deep learning algorithms can automatically identify features in raw data as the neural network learns. Taking word embeddings as input, Cocos et al. [15] applied a LSTM classifier to label the ADRs in tweets. They also investigated the effect of different word representations on the final performance. Gupta et al. [36] proposed a semi-supervised RNN model to address the labeled data scarcity problem. For unsupervised learning, a BiLSTM was used to predict the drug name with its context, and was also trained to predict ADRs with supervised learning in the tweet. Similarly, Lee et al. [37] proposed a semi-supervised CNN to automatically extract features for ADR labeling. Li et al. [38] proposed a joint BiLSTM model to extract adverse drug events between drug and disease entities, and their resident relations between bacteria and location entities from biomedical texts. Deep neural networks can also be extended by stacking multiple layers. One viable option is to stack a CRF on a BiLSTM layer, so that the BiLSTM-CRF model [16, 17] can learn both long dependency and constraints between labels.

In addition to word embeddings alone, several studies have incorporated extra embeddings, such as character embeddings [39, 40], POS embeddings [41], and some other embeddings [42]. To further improve detection performance, an attention mechanism was introduced to align the target token to its context words and to pay additional attention to the contexts which contributes more to the final detection result. The self-attention can interact with the tokens in the sequence, and force them to learn the dependency of the classifier on different parts of the sequence [18, 25, 43].

## Imbalanced learning

Machine learning for imbalanced data distribution poses a challenge for classification models since most of the learning algorithms used for classification implement an accuracy approximation function as the objective, i.e., cross entropy. These algorithms are designed around the assumption of equal examples numbers for each class. When the distribution of examples across the known classes is biased or skewed, the minority classes may be covered by the majority classes and thus will be ignored by the classifier, resulting in poor predictive performance, specifically for the minority class. Several previous works have sought to address the data imbalance issue at either the data or algorithm level.

**Data-level methods.** To address the data imbalance issue, several previous works applied either down-sampling [44] or up-sampling strategies [44, 45] to balance the data distribution and exploit the most difficult examples of the minority class. In their simplest form, a down-sampling strategy discards random samples from the majority classes while an up-sampling strategy duplicates from minority classes. Unfortunately, the former reduces the total amount of the training data while the latter provides a large number of repeated samples, which has been shown to cause over-fitting.

To strengthen class boundaries and reduce over-fitting, Chawla et al. [46] proposed the synthetic minority over-sampling technique (SMOTE) to produce artificial minority samples by the margin between existing minority samples and their nearest neighbors with the same labels. However, these methods are useless in sequence labeling tasks, since increasing or reducing the number of individual tokens in a text-sequence alone will result in the loss of the syntactic relations of the sentence.

**Algorithm-level methods.** Instead of handling the training data distribution, the algorithm-level methods consider a class penalty or weight to reduce bias towards the minority classes [47–50]. For cost-sensitive learning, penalties or weights are assigned to each class through a cost matrix. Obviously, increasing the cost of the minority samples is equivalent to increasing their importance or decreasing the likelihood that the model will incorrectly misclassify the sample.

Another line of data resampling is to applied alternative objective function instead of cross entropy to control the weights of examples in training procedures. For examples, focal loss [29] applies a modulating term to the cross entropy loss to focus learning on hard negative samples. It can automatically reduce the weight for easy samples and thus force the model focus on hard samples. Dice loss [30] applies an approximate function of $F$-measure to attach similar importance to false positives and false negatives. Boosting algorithms such as AdaBoost [51] select and weight hard examples to train classifiers.

## Conclusions

In this study, a weighted conditional random field is proposed for imbalanced-data ADR detection tasks. It applies a pre-trained language model as a context encoder which is both robust to strong class imbalanced datasets and can integrate dependencies and syntactical

information between tokens. To address the data imbalance issue, a weighted variant of the CRF loss function is implemented to assign more weight to the minority class, forcing the model to pay more attention to such classes to ensure effective detection. Furthermore, we introduce an explainable algorithm which provides a qualitative understanding between the input features and the corresponding prediction to compare the behaviors of the models with and without the weighted loss function.

Experimental results show that the proposed weighted variant of CRF provides a significant performance boost without changing the model architecture in imbalanced-data tasks. Future work will attempt to adjust the weights of training samples based on target metrics or to build a separate network for weight prediction.

## Author Contributions

**Conceptualization:** Jin Wang, Liang-Chih Yu.

**Formal analysis:** Jin Wang, Liang-Chih Yu.

**Supervision:** Xuejie Zhang.

**Writing – original draft:** Jin Wang.

**Writing – review & editing:** Liang-Chih Yu, Xuejie Zhang.

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
