## [Decision Letter · Decision Letter 0]

24 Feb 2022

Dear Prof. Yu,

Thank you very much for submitting your manuscript "Explainable Detection of Adverse Drug Reaction with Imbalanced Data Distribution" for consideration at PLOS Computational Biology.

As with all papers reviewed by the journal, your manuscript was reviewed by members of the editorial board and by several independent reviewers. In light of the reviews (below this email), we would like to invite the resubmission of a significantly-revised version that takes into account the reviewers' comments.

We cannot make any decision about publication until we have seen the revised manuscript and your response to the reviewers' comments. Your revised manuscript is also likely to be sent to reviewers for further evaluation.

Sincerely,

Andrey Rzhetsky

Associate Editor

PLOS Computational Biology

Mark Alber

Deputy Editor

PLOS Computational Biology

Reviewer's Responses to Questions

**Comments to the Authors:**

Reviewer #1: The authors propsed a new machine learning model for Adverse Drug Reaction detection, especially focused on the imbalance issue. They showed promising results based on this model. The reviewer appreciates the novelty of the work as well as the clarity of writing. But there are a few questions that need to be answered before it can be published:

1. While the key component of the proposed model is CRF, the authors have not explained CRF in detail.

2. Could you explain why the proposed model consist of an LSTM followed by the BERT module? To the reviewer's understanding, BERT itself is able to handle the sequential information.

3. Why the train/test splits are different for Tweets and Pubmed data?

4. Fig. 2a and 2c are confusing to the reviewer. Shouldn't the performance converge to a specific value as the epochs increase? If the model is overfitted with large epochs, you should add regularization/dropout to make it converge, instead of using an epoch that is not converged.

5. Equation 12 might be misleading, dash '-' is not distinguishable with minus sign.

Reviewer #2: **** Summary of the paper ****

The objective of this paper is to tackle the data imbalance issue in sequence labeling tasks. Instead of using the conventional softmax in the output layer, a weighted variant of Conditional Random Field (CRF) is proposed to capture the relationship of labels between tokens. Experiments show that the CRF layers improve the existing sequence modeling architectures in two Adverse Drug Reactions (ADR) tasks.

**** Main review ****

Strengths:

(1) I think the data imbalance problem in machine learning in general is interesting. This work focuses specifically on the sequence labeling tasks in Natural Language Processing (NLP). This work might have a greater impact if extended to address the data imbalance problem in other data types such as time-series (e.g., speech).

(2) The authors did a good job in summarizing imbalanced learning techniques in the related works and classified them into data-level methods and algorithm-level methods.

(3) Table 3 in the experiments is extensive. That includes comparison of using softmax or CRF in the output layer of several existing NLP architectures. The experiments did show that the weighted version of CRF slightly improves across 3 metrics (precision, recall and F1-score) on both tasks (Twitter and PubMed).

Weaknesses:

(1) I am not certain if this work fits into the theme of PLOS Computational Biology. Fundamentally, this work is purely NLP. My honest thought is other NLP-focused journals/conferences might be more suitable.

(2) After a quick check in the Internet, I found that the idea of putting CRF on top of existing NLP architectures is not new. To name a few:

- (Huang et al., 2015) Bidirectional LSTM-CRF Models for Sequence Tagging, https://arxiv.org/abs/1508.01991

- (Souza et al., 2019) Portuguese Named Entity Recognition using BERT-CRF, https://arxiv.org/abs/1909.10649

- PyTorch tutorial on a combination of Bi-LSTM and CRF: https://pytorch.org/tutorials/beginner/nlp/advanced_tutorial.html

(3) There is no mentioning that if the implementation/code will be released for reproducibility. What is the choice of deep learning framework to implement the models (e.g., PyTorch or TensorFlow or something else)?

**** Summary of the review ****

The data imbalance problem is meaningful. The results show that models with CRF on top outperform the ones with softmax. However, the novelty of the proposed method is not significant, because many aspects already existed in the literature. My recommendation leans towards rejection.

**** Correctness ****

All of the claims and statements are well-supported and correct.

**** Technical novelty and significance ****

The contributions are only marginally significant or novel. Aspects of the contributions exist in prior work.

**** Empirical novelty and significance ****

The contributions are only marginally significant or novel.

**** Recommendation ****

Marginally below the acceptance threshold. I am more leaning towards rejection.

**** Confidence ****

I am confident in my assessment, but not absolutely certain.

**Have the authors made all data and (if applicable) computational code underlying the findings in their manuscript fully available?**

Reviewer #1: Yes

Reviewer #2: **No: **The data used in experiments are publicly available, but the actual implementation/code is not.

PLOS authors have the option to publish the peer review history of their article (what does this mean?). If published, this will include your full peer review and any attached files.

Reviewer #1: No

Reviewer #2: **Yes: **Truong Son Hy
---

## [Decision Letter · Decision Letter 1]

26 Apr 2022

Dear Prof. Yu,

We are pleased to inform you that your manuscript 'Explainable Detection of Adverse Drug Reaction with Imbalanced Data Distribution' has been provisionally accepted for publication in PLOS Computational Biology.

Best regards,

Andrey Rzhetsky

Associate Editor

PLOS Computational Biology

Mark Alber

Deputy Editor

PLOS Computational Biology

Reviewer's Responses to Questions

**Comments to the Authors:**

Reviewer #1: The previous questions are well answered and the reviewer believes it is ready to publish

**Have the authors made all data and (if applicable) computational code underlying the findings in their manuscript fully available?**

Reviewer #1: Yes

PLOS authors have the option to publish the peer review history of their article (what does this mean?). If published, this will include your full peer review and any attached files.

Reviewer #1: No

---

## [Editor Report · Acceptance letter]

9 Jun 2022

PCOMPBIOL-D-21-02246R1 

Explainable Detection of Adverse Drug Reaction with Imbalanced Data Distribution

Dear Dr Yu,

I am pleased to inform you that your manuscript has been formally accepted for publication in PLOS Computational Biology. Your manuscript is now with our production department and you will be notified of the publication date in due course.

With kind regards,

Olena Szabo
